# A Route Planning Method for UAV Swarm Inspection of Roads Fusing Distributed Droneport Site Selection

**DOI:** 10.3390/s23208479

**Published:** 2023-10-15

**Authors:** Yingchun Zhong, Shenwen Ye, Yizhou Liu, Jingwen Li

**Affiliations:** 1Automation School, Guangdong University of Technology, Guangzhou 510006, China; gzzhw@126.com (Y.Z.); aieno@aliyun.com (S.Y.); 2Department of Computer Science, Sichuan Normal University, Chengdu 610101, China; 20231302008@stu.sicnu.edu.cn

**Keywords:** autonomous inspection, distributed droneport, site selection method, route planning, UAV

## Abstract

Current methods that use Unmanned Aerial Vehicle (UAV) swarms to inspect roads still have many limitations in practical applications, such as the lack of or difficulty in the route planning, the unbalanced utilization rate of the UAV swarm and the difficulty of the site selection for the distributed droneports. To solve the limitations, firstly, we construct the inspection map and remove the redundant information irrelevant to the road inspection. Secondly, we formulate both the route planning problem and the droneport site selection problem in a unified multi-objective optimization model. Thirdly, we redesign the encoding strategy, the updating rules and the decoding strategy of the particle swarm optimization method to effectively solve both the route planning problem and the droneport site selection problem. Finally, we introduce the comprehensive evaluation indicators to verify the effectiveness of the route planning and the droneport site selection. The experimental results show that (1) with the proposed method, the overlapped part of the optimized inspection routes is less than 7% of the total mileage, and the balanced utilization rate of the UAVs is above 75%; (2) the reuse rate of the distributed droneports is significantly improved after optimization; and (3) the proposed method outperforms the ant colony optimization (ACO) method in all evaluation indicators. To this end, the proposed method can effectively plan the inspection routes, balance the utilization of the UAVs and select the sites for the distributed droneports, which has great significance for a fully autonomous UAV swarm inspection system for road inspection.

## 1. Introduction

The condition of the road surface has essential impacts on transportation safety. The road administration needs to regularly detect and measure the road distress to maintain the road quality and enhance its safety [1]. Currently, road conditions are mainly monitored by manual inspection; however, such a strategy is both time-consuming and labor-costly [2]. Given the light weight, the high mobility and the low cost of the multirotor unmanned aerial vehicles (UAVs), using a UAV swarm to inspect roads is one of the main technologies in road detection [3]. However, current methods that use UAV swarms to inspect roads still have many limitations in practical applications, such as the lack of or difficulty in route planning, the unbalanced utilization rate of the UAV swarm and the difficulty of the site selection for a distributed droneport. Thus, route planning for the UAV swarm to inspect roads and the site selection for the distributed droneports are the keys to achieve fully autonomous UAV swarm inspection of roads.

The route planning of the UAV swarm mainly includes two categories, i.e., UAV swarm real-time route planning and UAV swarm offline route planning. The former refers to planning and updating the routes when the UAV swarm is working, and the latter refers to planning the complete routes before the UAV swarm moves. Pertaining to the UAV swarm real-time route planning, AI Duhayyim, M. et al. [4] applied the 6G UAV communication routing planning technology to enable the UAVs to efficiently collect data and plan the routes. Lee, M.-T. et al. [5] applied the edge computing system to provide the UAVs a higher level of autonomous control, which enabled the UAVs to be more flexible to automatically adapt to the dynamic changes of the environment. To optimize the route planning of the UAVs in the peak traffic period, Wang, K. et al. [6] introduced the concept of the team orienteering arc routing problem with time-varying profits and then proposed a route planning method that considered the spatiotemporal variations in monitoring demand. A bio-inspired route planning algorithm was leveraged in [7] to address the dynamic obstacle avoidance route planning problem for the UAVs when the environment maps are unknown. However, the literature [8] points out that drones are equipped with a limited battery supply and onboard computational power, which means they are unsuitable for detecting damage on roads in real-time. Similarly, conventional road inspection route planning is unsuitable for real-time calculation. Pertaining to the UAV swarm offline route planning, Cho, S-W. et al. [9] proposed a method that engenders a search path to cover all nodes with the minimum computation time for a fleet of heterogeneous UAVs. With the objective defined as minimizing both the number of UAVs and their total flying distance, Phalapanyakoon, K. and Siripongwutikorn, P. [10] studied the route planning problem with the mission time constraint and the battery capacity constraint, where each UAV may have more than one trip in a complete route due to the limited battery capacity. Considering the multi-regional route planning problem and multiple UAVs with the energy constraint, Xie, J. and Chen, J. [11] leveraged (1) a branch-and-bound based method to engender (near) optimal routes and (2) a genetic algorithm to efficiently engender routes for large-scale problems. Zhang, H. et al. [12] introduced a hybrid differential evolution algorithm to plan high-quality routes for fixed-wing UAVs in complex three-dimensional environments. Most of the existing studies only consider minimizing the flying distance of UAVs in the route planning process and ignore other goals such as the balanced utilization rate of the UAVs, where the corresponding droneport site selection problem is also unsolved.

Droneport site selection is an optimization decision-making problem that involves multiple factors. Current studies on the droneport site selection problem mainly focus on civil droneports. Alves, CJP. et al. [13] systematized the droneport siting criteria and proposed a site selection decision framework with a more objective decision-making process. Erkan, TE. et al. [14] utilized the geographic information system to select proper droneport sites, which considered 23 indicators and applied the analytic hierarchy process method and the rank-order centroid method to select the best droneport locations. Aydin, N. and Seker, S. [15] designed a guiding framework to select a hub droneport location within Turkey to satisfy demand and attract tourists. According to the bird ecological conservation data, Zhao, B. et al. [16] evaluated the impact of the different droneport site selection schemes on bird ecology and then selected a more suitable site for the sustainable development of humans and nature. An emergency droneport site selection was proposed in [17] based on the GeoSOT-3D global subdivision grid model, which verifies that the discrete global grid system has good suitability when performed as a spatial data structure for site selection. Liao, Y. and Bao, F. [18] introduced the fuzzy decision-making thesis to evaluate different droneport site selection methods, which defined the dominance degree among the methods for evaluation by using the difference between the two triangular fuzzy numbers. Due to the fact that civil droneports are seriously limited by the surrounding environment, most of the current studies mainly consider the noise, the ecology, the electromagnetic radiation and other factors when making decisions on selecting the droneport sites. Compared to the civil droneport, the UAV distributed droneports have looser requirements on the surrounding environment pertaining to the site selection, and more focus on the utilization rate of the droneport cluster and the inspection cycle. To our knowledge, there are few studies on the site selection of the UAV distributed droneport.

As the supply stations and maintenance stations for UAVs, the distributed droneports play a key role in the unmanned inspection system and the route planning of UAVs. In general, the solution quality of droneport site selection and route planning directly affect the performance of the unmanned inspection system. Thus, the good cooperation between them is the key challenge and opportunity to realize the full performance potential of unmanned inspection systems. To this end, this paper proposes a route planning method for the UAV swarm that fuses the site selection of the distributed droneports. Firstly, we construct the inspection map and remove the redundant information outside the target regions of the inspection. Secondly, we build a multi-objective optimization function, and formulate both the route planning problem and the droneport site selection problem in a unified problem. Thirdly, based on the particle swarm optimization method, we redesign the encoding strategy, the updating rules and the decoding strategy of the particle to effectively solve the problem of the route planning of the UAV swarm inspection and the droneport site selection.

## 2. Proposed Research Structure

The research structure of this paper is illustrated in Figure 1. The overall structure of the proposed method for solving both the route planning problem for the UAV swarm and the distributed droneport site selection problem includes four parts: the inspection map construction, the route inspection model formulation, the redesign of the particle encoding and decoding method and the experiments with comparison and analysis.

Pertaining to the inspection map construction, we first collect the map of the inspection target regions, and then separate the layers of the target regions to remove the redundant information outside those regions. Then, we set multiple candidate points (i.e., the candidate sites) for the droneport, simulate the inspection roads and divide the inspection roads and nodes into different parts, and simplify the complex roads in the target regions to provide the candidate droneport sites. Pertaining to the mathematical model for both the route planning problem and the droneport site selection problem, we build the objective function and constraints based on the practical demands of the road inspection to obtain an inspection optimization model under a specific number of UAVs. Pertaining to the algorithm, we improve the encoding method and the updating rules of the particles and design a corresponding particle decoding strategy. By doing so, the proposed algorithm enables each single particle to perform both the route planning and the droneport site selection, which further contributes to find the optimized route planning scheme and the droneport site selection scheme under a specific number of UAVs. Pertaining to the experiments and the analysis, we propose four indicators to evaluate the effectiveness of the route planning method and the droneport site selection method, and further compare our method with the ant colony optimization algorithm (ACO).

## 3. Inspection Map Construction

This paper takes the Panyu district (Guangzhou City, Guangdong Province, China) as an example to show the process of solving both the route planning problem and the droneport site selection problem. In the context of Panyu district’s cartographic preparations for the strategic establishment of prospective droneports, it becomes evident that the existing array of Geographic Information System (GIS) software, although diverse and capable, fails to comprehensively align with the precise demands delineated within the scope of this scholarly endeavor. Thus, we undertake a meticulous and deliberate endeavor in the manual crafting of a cartographic representation, meticulously tailored to the intricacies and prerequisites inherent to the droneport layout envisaged for Panyu district. This bespoke approach contributes to the meticulously calibrated and customized map for the exacting specifications mandated by the exigencies of our research inquiry.

### 3.1. Separation of the Layers of the Target Regions

We construct the inspection map of the Panyu district as shown in Figure 2, where the collected (i.e., original) map of this district is illustrated in Figure 2a, and the abundant part in Figure 2a is cleared manually as shown in Figure 2b.

### 3.2. Settings of the Candidate Points for Droneport

Since the network complexity of the provincial and county roads is much higher than that of the expressway networks, this paper mainly focuses on the provincial and county roads. Firstly, we need to settle the distance between distributed droneport candidate points along each road on the manually processed map of the Panyu district as the candidate points for the droneport as shown in Figure 2c. Given that the relatively economical flight speed of the UAV is 50 km/h and the maximum flight time is about 55 min, a distance greater than 5 km between two candidates can seriously affect flight safety, while a distance lower than 1 km may lead to large amount of distributed droneport candidates, and further, huge computation. Thus, we set such a distance to 3 km to achieve a good balance between the flight safety and reasonable computation by using the U880 UAV of Guangzhou Youfei Information Technology Co., Ltd. (Guangzhou, China) for conducting road inspection in experiments. Then, we collect the coordinates of all the candidate points and present the serial number for them as shown in Figure 2d. The numbers on the horizontal and the vertical axes refer to the corresponding coordinates, and the 1 unit on the coordinate axis represents 2.22 km. With the upper left corner taken as the origin node, the positions of the candidate points are represented by the corresponding horizontal and vertical coordinates. 

### 3.3. Simulation of the Inspection Roads

Based on Figure 2d, we illustrate the polyline segments to connect the candidate points along the road so as to simulate the inspection roads as shown in Figure 2e. Specifically, the total length of the inspection roads of the Panyu district is 888.19 km. 

### 3.4. Division of the Inspection Roads and the Candidate Points

As shown in Figure 2f, to simplify the model, we take each road fork and each road end as two endpoints, which are highlighted in black color, and take all the polyline segments and the passing nodes between such two endpoints as a whole inspection road. By doing so, we not only efficiently divide the inspection roads and the nodes, but also obtain the necessary routes for UAVs.

## 4. The Mathematical Model

### 4.1. Objective Function

There are two objectives for UAVs to complete a global road inspection, i.e., (1) achieving shorter inspection mileage of all UAVs, which could increase the inspection efficiency and decrease the energy cost; (2) achieving a more balanced utilization of UAVs by minimizing the difference of the mileages of all UAVs in a single inspection cycle, which avoids excessive maintenance since UAVs need to be regularly maintained and replaced. To achieve this, we set the objective function as both minimizing the inspection distance and maximizing the UAVs’ utilization rate during a global inspection cycle. To be specific, the global inspection cycle refers to a manual set time that is required to complete the inspection of all roads in a target region. In this paper, the global inspection cycle of the Panyu district is set to 22 working days per month.

#### 4.1.1. Total Mileage of UAVs

(1)$$Dis=\sum_{i}^{n}dis_{i},$$
where the Dis is the total flying mileage of all the UAVs, $dis_{i}$ is the flying mileage of the *i*-th UAV and *n* is the number of the UAVs in a global inspection cycle. Specially, the $dis_{i}$ is defined as follows:(2)$${dis}_{i}=M_{scale}\sum \sqrt{{\left(x_{j}-x_{j+1}\right)}^{2}+{\left(y_{j}-y_{j+1}\right)}^{2}},$$
where $\left(x_{j},y_{j}\right)$ and $\left(x_{j+1},y_{j+1}\right)$ are the coordinates of each two adjacent droneport candidate points in the inspection roads of the *i*-th UAV, and $M_{scale}$ transforms the distance calculated in the inspection map into the practical mileage. $M_{scale}$ is set to 2.22 km since 1 unit in the axises represents 2.22 km.

#### 4.1.2. UAVs’ Utilization Rate

We represent the UAVs’ utilization rate as the variance of the total mileages of all UAVs. In an inspection cycle, a smaller variance in the mileages of UAVs indicates a smaller difference in the mileage between two UAVs, which shows a higher utilization rate. Specifically, the variance in the total mileages of all UAVs is defined as follows:(3)$$\sigma^{2}=\frac{1}{n}\sum_{i}^{n}{\left(dis_{i}-\bar{dis}\right)}^{2},$$
where $\bar{dis}$ is the average value of the $dis_{i}$, which is shown as
(4)$$\bar{dis}=\frac{1}{n}\sum_{i}^{n}dis_{i}.$$

### 4.2. Constraints

#### 4.2.1. Distances between Adjacent Droneport Candidate Points

In this paper, the UAV with model No. U880 is employed to perform the route inspection, which achieves an average speed of 50 km per hour and a cruise duration of 55 min and is a typical product from the Guangzhou Youfei Information Technology Co., Ltd. Based on these features, we set the flying speed of the UAVs to 50 km per hour and the flying time to 30 min during the UAV inspection process, where the UAVs fly about 30 km for each journey. Regarding such a setting, the UAVs usually need to reserve about 30–40% of their power to deal with emergency accidents (e.g., continuous hovering in the air after losing contact, stable and nearby landing when the battery is low) in the real-world inspection process. Thus, we set the inspection time of the UAVs to about 30 min. Accordingly, for the UAVs, flying 30 min can roughly support a flight distance of about 30 km. Additionally, the practical flying mileage might be less than 30 km if we further consider the factors such as tailwinds and headwinds. Moreover, given that the spacing between the adjacent droneport candidate points in all routes is set to 3 km, we constrain the maximum distance between adjacent droneport candidate points in each route to 30 km and the minimum distance to 27 km. The constraint is formulated as follows:(5)$$27\;\mathrm{km}\leq D_{ab}\leq 30\;\mathrm{km},$$
where $D_{ab}$ refers to the distance between the adjacent droneport *a* and droneport *b*. Specifically, the $D_{ab}$ is defined as follows:(6)$$D_{ab}=M_{scale}\sum \sqrt{{\left(x_{k}-x_{k+1}\right)}^{2}+{\left(y_{k}-y_{k+1}\right)}^{2}},$$
where $\left(x_{k},y_{k}\right)$ and $\left(x_{k+1},y_{k+1}\right)$ refer to the horizontal and vertical coordinates of adjacent droneport candidate points in the route segment between the droneport *a* and the droneport *b*.

#### 4.2.2. Overlapped Rate of the Inspection Routes

Given the inspection routes of a UAV, the overlapped rate of the routes represents the proportion of the overlapped mileage in the total mileage of the routes. Let $d_{r}^{i}$ denote the overlapped mileage in the inspection routes of the *i*-th UAV and $\rho_{i}$ denote the overlapped rate of the inspection routes of the *i*-th UAV. We define the $d_{r}^{i}$ and $\rho_{i}$ as follows:(7)$$d_{r}^{i}=M_{scale}\sum \sqrt{{\left(x_{u}-x_{u+1}\right)}^{2}+{\left(y_{u}-y_{u+1}\right)}^{2}},$$
(8)$$\rho_{i}=\frac{d_{r}^{i}}{dis_{i}}\times 100\%,$$
where $\left(x_{u},y_{u}\right)$ and $\left(x_{u+1},y_{u+1}\right)$ refer to the coordinates of adjacent droneport candidate points in the overlapped part of the inspection routes of the *i*-th UAV. Given that the lower overlapped rate of the inspection routes means the higher efficiency of road inspection, it has been duly noted that contemporary path-planning algorithms encounter a persistent challenge pertaining to the occurrence of redundant inspection routes as mentioned in [19]. Specifically, the efficacy of path coverage hovers at approximately 83%, thereby signifying that a substantial 17% proportion of the path coverage comprises duplicate inspection trajectories. In order to expedite algorithmic search processes, we advocate for a nuanced relaxation of the established constraint parameters, manifesting in the deliberate capping of inspection route redundancy at an upper limit of 18%. Thus, the $\rho_{i}$ is limited with a maximum,
(9)$$\rho_{i}\leq 0.18,$$
which decreases the overlapped rate of the inspection routes to the greatest extent so as to improve the solution quality.

#### 4.2.3. Flying Time of UAVs

The UAVs perform the road inspection by flying from one droneport to an adjacent droneport. In such a process, except for the time that is required in the road inspection, we also need to reserve the time intervals for other tasks such as the maintenance of UAVs as shown in Table 1.

Thus, the maximum time for UAVs to complete a road inspection task could be 3 h and 10 min, i.e., 3.17 h. To this end, we define the inspection time of the route between the adjacent droneport *a* and droneport *b* for the *i*-th UAV as $t_{ab}^{i}$ and constrain it to less than 3.17 h as below,
(10)$$t_{ab}^{i}\leq 3.17\;h.$$

**Remark 1.** 
*This paper adopts the charging method for UAVs due to the high failure rate of replacing the battery of UAVs in the droneport.*


#### 4.2.4. Time Period for Performing a Global Inspection

In the road inspection process, we denote the starting position of the inspection routes and UAVs as the droneport. The UAVs perform the inspection task twice a day, i.e., 9:00 a.m. and 14:00 p.m., until completing a global inspection. In this paper, we define the time period of a global inspection *T* as the maximum inspection time among the inspection routes as follows:(11)$$T=max\left\{T_{s}^{1},T_{s}^{2},\cdots ,T_{s}^{n}\right\},$$
where $T_{s}^{i}=\frac{N_{i}-1}{2}$ is the inspection time of the *i*-th UAV and $N_{i}$ is the number of droneports that the *i*-th UAV passes during the whole inspection process. Thus, the number of times that the *i*-th UAV performs the inspection task is $N_{i}-1$. Note that the unit of $T_{s}^{i}$ is “days”.

#### 4.2.5. Candidate Droneport Nodes Outside the Necessary Routes

Given that the planned routes for UAVs sometimes distribute outside the necessary routes, the UAVs may stop and need to be charged at such non-necessary regions. To satisfy this demand, we set a candidate droneport outside the necessary route with the serial number $num_{e}$ that is defined as below,
(12)$${num}_{e}={num}_{max}+1,$$
where the $num_{max}$ is the largest serial number in the existing candidate droneport.

The position of the candidate droneport with serial number $num_{e}$ could be inferred in Equations (Equation 13) and (Equation 14),
(13)$$x_{e}=\left\{\begin{array}{cc} x_{ago}+\frac{dis_{ration}-dis_{ago}}{\sqrt{1+k_{l}^{2}}} & \mathrm{if}\;x_{ago}\leq x_{later}\\ x_{ago}-\frac{dis_{ration}-dis_{ago}}{\sqrt{1+k_{l}^{2}}} & \mathrm{if}\;x_{ago}>x_{later} \end{array}\right.$$
(14)$$y_{e}=k_{l}\left(x_{e}-x_{ago}\right)+y_{ago},$$
(15)$$dis_{ration}=\frac{30}{M_{scale}}=13.5\;\mathrm{cm},$$
where $\left(x_{e},y_{e}\right)$ refers to the coordinates of this droneport, $\left(x_{ago},y_{ago}\right)$ refers to the coordinates of the end candidate droneport of the last inspection route and $x_{later}$ refers to the horizon coordinate of the starting droneport of the next inspection route. In Equation (Equation 15), we transform the practical 30 km into the distance in the map, which indicates that the distance between the candidate droneport outside the necessary routes and the last passed candidate droneport is set to 30 km. This strategy could avoid setting too many redundant droneports on non-necessary routes given enough distance between the droneports, which further saves on the construction cost of the droneports. Moreover, $dis_{ago}$ refers to the mileage of the last inspection route in the map, which is computed as follows:(16)$${dis}_{ago}=\sum \sqrt{{\left(x_{p}-x_{p+1}\right)}^{2}+{\left(y_{p}-y_{p+1}\right)}^{2}},$$
where $\left(x_{p},y_{p}\right)$ and $\left(x_{p+1},y_{p+1}\right)$ refer to the coordinates of adjacent droneport candidate points in the last inspection route.

Let $k_{l}$ denote the slope of the segment that is formed by the candidate droneport outside the necessary routes and the candidate droneport at the end of the last inspection route. We define the slope as
(17)$$k_{l}=\frac{y_{later}-y_{ago}}{x_{later}-x_{ago}},$$
where $y_{later}$ is vertical coordinate of the starting droneport of the next inspection route.

### 4.3. The Mathematical Model

Based on the objective functions and the constraints introduced defined above, the mathematical model of the proposed problem is formulated as follows:(18)$$\left\{\begin{array}{c} \left\{\begin{array}{c} min\;Dis=\sum_{i}^{n}dis_{i}\\ min\;\sigma^{2}=\frac{1}{n}\sum_{i}^{n}{\left(dis_{i}-\bar{dis}\right)}^{2} \end{array}\right.\\ s.t.\left\{\begin{array}{c} 27\;\mathrm{km}\leq D_{ab}\leq 30\;\mathrm{km}\\ \rho_{i}\leq 0.18\\ t_{ab}^{i}=3.17\;h\\ T=max\left\{T_{s}^{1},T_{s}^{2},\cdots ,T_{s}^{n}\right\}\\ num_{e}=num_{max}+1 \end{array}\right. \end{array}\right.$$

Specifically, the first objective function aims to minimize the total mileage of UAVs during a global inspection cycle, and the second objective function aims to minimize the difference of the utilization rate among UAVs so as to make the utilization rate of the UAV swarm more balanced. Regarding the constraints, the first one means that the distances between adjacent droneport candidate points are in the range of 27 km to 30 km. The second constraint means that the overlapped rate of the inspection routes cannot exceed 0.18. The third constraint means that the maximum time for UAVs to complete a road inspection task cannot exceed 3.17 h. The fourth constraint means that the time period of a global inspection is defined as the maximum inspection time among the inspection routes. The fifth constraint means that for a candidate droneport outside the necessary routes, its serial number is 1 greater than the existing maximum serial number.

## 5. The Proposed Algorithm

The conventional particle swarm algorithms usually generate a single route for each particle for solving the route planning problem. However, this design has difficulty handling the scenario where each particle needs to plan the routes for the UAV swarm and engenders multiple routes simultaneously. Thus, we propose a novel particle swarm algorithm with a new design for the encoding, updating and decoding strategies for solving both the route planning problem and the droneport site selection problem.

### 5.1. Particle Encoding Method

Given a specific number of UAVs, each UAV randomly selects a candidate droneport among all the roads as its starting position and inspects an uninspected road closest to that UAV after each inspection of the road. The inspection process is repeated until all the roads are inspected (i.e., completing a global inspection). To this end, a solution for the global inspection with multiple routes is engendered for each UAV. In this paper, we redesign the particle encoding method to enable each particle to generate a solution for the UAV swarm by fusing the solutions of all UAVs together.

With the route represented as a sequential order of the candidate droneports, e.g., 3→5→4→⋯, the routes for *n* UAVs include *n* sequences of the candidate droneports. We combine these *n* sequences together into a single one to retrieve the particle encoding as illustrated in Figure 3, where bp is the breakpoint to separate the routes of different UAVs.

### 5.2. The Updating Rules

To enable the particles to randomly plan the routes and select the droneport sites for a broader search range, we propose the particle updating rules to update the information of the particle swarm.

(1) Subtracting the position Θ: The speed between two positions equals their subtraction. For example, given two positions *x* and *y*, the corresponding speed is defined as below,
(19)xΘy=y→x

(2) Adding the speed ⊕: The new speed is obtained by summing up two speeds. For example, given two speeds v=a→b,w=x→y, the new speed is defined as below,
(20)$$v⊕w=\left(a\to b\right)⊕\left(x\to y\right)=\left\{\begin{array}{cc} x\to y & \mathrm{if}\;\;x\neq y\\ a\to b & \mathrm{if}\;\;x=y \end{array}\right.$$

(3) Adding the position and the speed ⊕: The new position could be retrieved by adding a position and a speed. For example, given a position *a* and a speed v=x→y, the new position is updated as below,
(21)$$a⊕v=a⊕\left(x\to y\right)=\left\{\begin{array}{cc} rand & \mathrm{if}\;\;a\neq x\\ y & \mathrm{if}\;\;a=x \end{array}\right.$$

(4) Speed multiplier ⊗: The new speed is computed by multiplying a coefficient and a speed. For example, given a random coefficient c∈[0,1] and a speed v=x→y, the new speed is updated as below,
(22)$$c⊗v=c⊗\left(x\to y\right)=\left\{\begin{array}{cc} x\to y & \mathrm{if}\;\;c>d\\ x\to x & \mathrm{if}\;\;c\leq d \end{array}\right.$$
where d∈(0,1) is a random number. The positions and the speeds of the particles are updated as below,
(23)$$v_{id}^{t+1}=\omega ⊗v_{id}^{t}⊕c_{1}⊗\left(p_{id}^{t}\Theta x_{id}^{t}\right)⊕c_{2}⊗\left(p_{gd}^{t}\Theta x_{id}^{t}\right)$$
(24)$$x_{id}^{t+1}=x_{id}^{t}⊕v_{id}^{t+1}$$

Based on the definitions and the rules mentioned above, the position of the particles could be updated using Equation (Equation 25) regardless of the value of the inertia weight ω introduced in [20].
(25)$$\;x^{k+1}\;=\;\left\{\begin{array}{cc} \;P_{gd}^{k} & \mathrm{if}\;\mathrm{rand}(0,1)<c_{2}\\ \;P_{id}^{k} & \mathrm{if}\;c_{2}\leq \mathrm{rand}(0,1)<c_{1}\\ \;\mathrm{rand} & \mathrm{if}\;\mathrm{rand}(0,1)\geq c_{2},\;\mathrm{rand}(0,1)\geq c_{1} \end{array}\right.$$
where $x^{k+1}$ is the updated position of the particle, $P_{id}^{k}$ is the historical best solution of the particle *i*, $P_{gd}^{k}$ is the optimized solution of the particle swarm and rand(0,1)∈ [0,1] is a random number [20,21]. Specifically, $c_{1}$ represents the learning factor for individual particles, and a larger value of $c_{1}$ can lead to particles becoming trapped in local optima. Conversely, $c_{2}$ refers to the learning factor for the entire particle swarm, and a larger value of $c_{2}$ can make it more difficult for the search process to converge. To achieve an exquisite balance between the proclivities for local and global exploration inherent to our particle swarm, we apportion the capacities for local search, global search, and stochastic exploration in a carefully calibrated ratio of 1:1:3. Consequently, we arrive at the meticulously chosen values of $c_{1}=0.4$ and $c_{2}=0.2$.

### 5.3. Decoding Method

#### 5.3.1. Direct Decoding Strategy

The direct decoding strategy aims to extract the planned routes for UAVs from the particle encoding information. As designed in Section 6.1, the particle encoding is combined with multiple solutions of UAVs with some breakpoints to differentiate them. Thus, the encoding information could be directly decoded to retrieve the planned routes of UAVs by separating the particle encoding according to the breakpoints as shown in Figure 4.

#### 5.3.2. Indirect Decoding Strategy

The indirect decoding strategy aims to extract the selected droneport sites from the particle encoding. Based on the abovementioned planned routes, the indirect decoding strategy first sets an droneport at each starting position of the routes, then selects the candidate droneport nodes that satisfy all the constraints of the distance between the adjacent candidate droneport on each route as the final droneport sites. As shown in Figure 5, $D_{i}$ is the distance satisfying the constraints, and the numbers marked in the red color are the serial numbers of the candidate droneport selected as the droneport sites.

## 6. Experiments

We conduct all experiments on a Windows7 operating system, where the proposed algorithm is programmed using Python and all experiments are conducted in a Intel (R) Core (TM) I5-7300HQ CPU @ 2.50 GHz, 8 GB RAM.

### 6.1. Evaluation Indicators

#### 6.1.1. Balanced Utilization Rate

The balanced utilization rate β is defined as the ratio of the shortest mileage to the longest mileage during a global inspection cycle as follows:(26)$$\beta =\frac{min\left\{dis_{1},dis_{2},\cdots ,dis_{n}\right\}}{max\left\{dis_{1},dis_{2},\cdots ,dis_{n}\right\}}\times 100\%$$

Specifically, β closing to 1 indicates small differences among the UAV mileages and a highly balanced utilization rate. Similarly, β much smaller than 1 indicates that the shortest mileage is much smaller than the longest one, which shows a less balanced utilization rate. Specifically, this indicator serves as a fundamental indicator of equilibrium within the domain of UAV deployment. Proximity to a value of 1 within this metric is indicative of a heightened state of equilibrium in UAV utilization, denoting a state where resources are distributed in a highly balanced manner across the fleet of drones.

#### 6.1.2. Optimization Rate of the Total Mileage of the UAV Routes

We define the optimization rate of the total mileage of the UAV routes $\eta_{Dis}$ as the ratio of the mileage of the inspection roads to the total mileage of the UAV routes during a global inspection cycle, which is formulated as follows:(27)$$\eta_{Dis}=\frac{{length}_{path}}{Dis}\times 100\%$$
where $length_{path}$ is the total mileage that needs the inspection. Taking the Panyu district as an example, the length of the roads that needs to be inspected is 888.19 km. Note that $\eta_{Dis}$ closing to 1 indicates that the total mileage of the UAV routes is relatively short in a global inspection cycle and the optimization rate of the total mileage of the UAV routes is relatively high. The essence of this indicator resides in elucidating the variability observed in the cumulative mileage of planned routes following each algorithmic iteration. A reduced degree of variability attests to the method delineated within this paper, manifesting a heightened level of congruity in the determination of total route mileage, thereby mitigating the likelihood of extreme scenarios.

#### 6.1.3. Relative Droneport Utilization Rate

Given a droneport site selection scheme where the number of droneports is α times the number of the UAVs, the relative droneport utilization rate $\gamma_{\alpha }$ is defined as the ratio of the total number of droneports on the routes to the total number of practical droneports, which is formulated as follows:(28)$$\gamma_{\alpha }=\frac{\sum N_{i}}{N}$$
where *N* is the total number of practical droneports and $\alpha =\frac{N}{n}$. Note that some droneports might be used multiple times in the practical site selection process, which means there are some droneports that may share the same location among the routes. Moreover, the total number of droneports on the routes could infer the total number of droneports that are necessary without the overlapping locations. Thus, based on the ratio $\gamma_{\alpha }$, we could retrieve the utilization rate of the droneports given a specific number of UAVs where the site selection is not coincident. Specifically, the higher value of $\gamma_{\alpha }$ indicates the higher utilization rate of droneports. The intrinsic importance of this indicator is rooted in its capacity to explicate the extent of repeated utilization within the framework of distributed droneports. As the frequency of this reuse surges, it unveils a concurrent decrease in construction expenditures, thereby accentuating the economic reverberations entwined with this phenomenon.

#### 6.1.4. Optimization Rate of a Global Inspection Cycle

The optimization rate of a global inspection cycle $\eta_{T}$ is defined as
(29)$$\eta_{T}=\left(1-\frac{T}{22}\right)\times 100\%$$

In this paper, we constrain a global inspection cycle within one month and limit the UAVs to inspecting during the working day. In such a setting, we use 22 to represent the number of working days within a month. Specifically, if the $\eta_{T}$ is closer to 1, a global inspection cycle is shorter, and the optimization rate of a global inspection cycle is higher. The inherent significance of this indicator lies in its capacity to elucidate the interplay between the quantity of unmanned aerial vehicles (UAVs) and the formulation of route planning strategies. The envisioned outcome posits that as this indicator tends toward stabilization, it serves as an indicator of the method outlined in this paper, achieving a state of consistent and reproducible route planning outcomes.

### 6.2. Experimental Design

We take the Panyu district as an example to verify the feasibility and the effectiveness of the proposed method. We set the initial population to 100, and the values of learning factors $c_{1}$ and $c_{2}$ to 0.4 and 0.2, respectively.

The experiments are designed to check the feasibility (i.e., Experiment 1) and the effectiveness (i.e., Experiments 2–6) of the proposed method.

Experiment 1: We verify the feasibility of a method by checking whether it can generate reasonable routes and droneport sites. If yes, the method is deemed to be feasible.

Experiment 2: We show the correlation between the balanced utilization rate of UAVs and the total number of UAVs. The primary objective of this experiment is to meticulously examine the influence of the aggregate quantity of UAVs on the equilibrium of UAV utilization. In particular, a notable decline in the equilibrium of utilization would signify potential challenges in sustaining a uniform performance, casting a discerning light on the viability of our proposed method.

Experiment 3: We show the correlation between the optimization rate of the total mileage of the UAV routes and the total number of the UAVs. The primary objective of this experiment is to investigate whether there is a significant correlation between the total number of UAVs and the total planned route mileage. The rationale behind this investigation is that maintaining a stable total route mileage can facilitate a more consistent maintenance cycle for the UAV fleet, leading to a reduction in management costs.

Experiment 4: We show the correlation of the utilization rate of the droneports and both the number of the UAVs and the number of the droneports. The primary objective of this experiment is to investigate the impact of changes in the number of UAVs and droneports on the reusable rate of distributed droneports. The reusable rate is a crucial metric that can significantly reduce construction costs, and its increase can have a profound impact on the scalability and sustainability of distributed droneport systems.

Experiment 5: We explore the optimization rate of a global inspection cycle and show the optimization rates for different inspection solutions. The primary objective of this experiment is to investigate the impact of changes in the number of UAVs on the reusable rate of distributed droneports. The reusable rate is a crucial metric that can significantly reduce construction costs, and its increase can have a profound impact on the scalability and sustainability of distributed droneport systems.

Experiment 6: We compare our strategy with the ACO algorithm to show the effectiveness of the proposed method. The primary objective of this experiment is to rigorously evaluate the efficacy of the proposed enhancements to the particle swarm optimization algorithm in this paper by conducting a comprehensive comparative analysis with other state-of-the-art optimization algorithms. The results of this analysis will provide valuable insights into the strengths and weaknesses of the proposed approach and its potential for practical applications.

### 6.3. Experimental Results and Analysis

#### 6.3.1. Experiment 1

We first retrieve the calculations carried out in the case of 5, 10 and 20 UAVs and the corresponding best particles. Then, we illustrate the route planning results and the droneport site selection results of the UAVs, which are obtained by decoding the best particles with different numbers of UAVs in Figure 6 and Figure 7, respectively.

In Figure 6, the green lines refer to the routes that need the inspection, the red lines refer to the overlapped inspection routes in a global inspection cycle and the blue lines refer to the UAV routes planned by the proposed algorithm. We can find that the planned UAV routes could cover all the routes in the region, which demonstrates that the proposed method can plan reasonable routes for the global inspection. Thus, we conclude that the proposed method is feasible for route planning. In Figure 7, the green lines refer to the routes that need the inspection, where the lines cross in blue refer to the droneports that are used only once during a global inspection cycle and the red line crossings refer to the droneports that are used multiple times during a global inspection cycle. We can observe that the selected droneport sites are all near the planned routes, and some droneports are repeatedly used for different numbers of UAVs, which reduces the number of droneports and further saves the construction costs. The observations demonstrate that the proposed method could select reasonable droneport sites and show the feasibility of our method in droneport site selection.

#### 6.3.2. Experiment 2

We first explore the optimization problem in the case of 1 to 20 UAVs and obtain the corresponding best particles, and then plan the routes by decoding those particles. Furthermore, we illustrate the balanced utilization rate of UAVs under different numbers of the UAVs based on the planned routes in Figure 8.

From Figure 8, we could observe that the UAV utilization rate gradually decreases as the number of UAVs increases. Specifically, the utilization rate finally fluctuates around 76% and tends to be stable with a minimum of 73.74%. We can conclude that based on the proposed method, the route planning schemes under different numbers of UAVs achieve satisfactory balanced utilization rates. This result demonstrates that as the number of the drones increases, the balanced utilization rate of UAVs exhibits no significant decrease. This resilience in maintaining a robust level of optimization underscores the effectiveness and durability of our proposed method.

#### 6.3.3. Experiment 3

Based on 20 types of the planned routes obtained from Experiment 2, we compute and collect the optimization rate of the total mileage of the routes under different number of UAVs. Then we illustrate the optimization rate under different numbers of UAVs in Figure 9.

Figure 9 shows that the optimization rate of the total mileage of the UAV routes slightly decreases as the number of UAVs increases, where the optimization rate fluctuates around 76%. We could conclude that under different numbers of UAVs, the proposed method plans the routes with a consistent and promising optimization effect on the total mileage of the routes. This result elucidates that, when employing our route planning method, the cumulative mileage of the routes maintains a remarkable level of stability even as the number of drones escalates. This steadfastness in route mileage holds the potential to foster a consistent maintenance cycle for the UAV fleet, consequently yielding a reduction in overall management costs.

#### 6.3.4. Experiment 4

Based on the 20 best particles found in Experiment 2, we decode these particles and obtain the selected droneport sites under different numbers of UAVs. We further compute the relative droneport utilization rate under different numbers of UAVs based on the selected droneport sites. Then, we illustrate the relative droneport utilization rate under different numbers of UAVs in Figure 10.

From Figure 10, we observe that the relative droneport utilization rate gradually increases as the number of UAVs increases and the former is always larger than 1. This indicates that the proposed method could select the droneport sites with a high relative droneport utilization rate and high profits. Moreover, there are always some repeatedly used droneports that are used multiple times during a global inspection cycle regardless of the number of UAVs, and the number of the repeatedly used droneports increases as the number of UAVs increases. Specifically, the highest relative droneport utilization rate is 1.52, which means that more than half of the droneports are repeatedly used. Thus, we can conclude that the proposed method could select the droneport sites with a relatively high droneport utilization rate. This result delineates a discernible correlation wherein the proliferation of UAVs precipitates a concomitant expansion in droneport facilities, notably marked by an augmented utilization of distributed airports. This notable trend underscores the potential for substantial reductions in construction expenses, thus exemplifying a compelling cost-saving facet within the realm of UAV infrastructure development.

#### 6.3.5. Experiment 5

Based on 20 route planning schemes obtained from Experiment 2, we compute the optimization rate of a global inspection cycle under different numbers of UAVs. Then, we illustrate the optimization rate of a global inspection cycle under different numbers of UAVs in Figure 11.

From Figure 11, we can observe that the optimization rate of a global inspection cycle increases with a maximum of 93.18% as the number of UAVs increases. Specifically, when the number of the UAVs exceeds 4, the optimization rate is already above 75%, which shows a relatively high optimization rate. With more UAVs, the increasing speed of the optimization rate significantly slows down. When the number of the UAVs exceeds 9, the optimization rate achieves 88.64%, where the optimization rate after that is only slightly increased. Thus, we can conclude that the proposed method could plan the routes with a satisfactory optimization rate for a global inspection cycle. This result illustrates that as the number of drones increases, the solution space for flight path planning expands, potentially leading to better flight path planning outcomes. However, beyond 9 drones, further increasing the number of drones does not significantly enhance the effectiveness of flight path planning. This observation underscores the nuanced dynamics that govern the interplay between UAV quantity and route optimization, revealing a point of diminishing returns in the pursuit of route planning refinement.

#### 6.3.6. Experiment 6

For ACO, the initial population is set to 100 and the pheromone weight is set to 1. The visibility weight is set to 2 and and the pheromone volatilization rate is set to 0.5. We apply ACO for solving the optimization problem with 13 UAVs, and obtain the route planning scheme, droneport site selection scheme and the optimization time. Based on the planned routes and selected droneport sites, the balanced utilization rate of UAVs, the optimization rate of the total mileage of the routes, the relative droneport utilization rate and the optimization rate of a global inspection cycle are computed. We summarize the results of ACO and our method in Table 2.

From Table 2, we can observe that pertaining to the optimization time, our method is faster than ACO, where the computation time of our method is about $\frac{1}{3}$ shorter than the ACO. Pertaining to the balanced utilization rate of UAVs, the rate of our method is more than 90%, while the rate of ACO is less than 70%, which is much lower than ours. Pertaining to the optimization rate of the total mileage of the routes, our method is superior to ACO, where the optimization rate of our method is more than 12% higher than that of ACO. Pertaining to the relative droneport utilization rate and the optimization rate of a global inspection cycle, our method also slightly outperforms ACO, which indicates that our method achieves a higher optimization rate of the road inspection and the droneport site selection than ACO. Based on the observations, we could conclude that our method outperforms ACO on all perspectives. Additionally, the time consumed by offline route planning before take-off of a swarm of delivery drones in the literature [22] ranges from 20 to 2173 s. By comparison, the time consumed by the method proposed in this paper is acceptable.

### 6.4. Discussion

Using a UAV swarm to perform inspection tasks is one of the future research trends. Following the delineation of targeted inspection areas, the foremost task at hand is to determine the locations and quantities of distributed droneports. Subsequently, leveraging this foundational information and considering the prevailing road network infrastructure, we embark on the meticulous planning of daily inspection routes for our fleet of UAVs. While extant literature has presented a plethora of studies concerning real-time route planning for UAV fleets [4,5,6,7] and offline planning methodologies [8,9,10,11], these endeavors, regrettably, have failed to incorporate the judicious selection of distributed droneport sites as an optimization objective. Moreover, they have largely neglected the critical issue of achieving equilibrium in UAV utilization. This prevailing oversight not only obfuscates the determination of optimal fleet sizes and the spatial distribution of distributed droneports but also exacerbates the problem of imbalanced drone utilization. Consequently, this paper posits the imperative of investigating the optimization goals encompassing the selection of distributed droneport locations alongside the equitable utilization of UAV resources. The experimental results in this paper show the following: (1) From Figure 8 and Figure 9, we can find that when the number of UAVs exceeds 14, the balanced utilization rate of UAVs is stable around 75%, which means the increase in the number of UAVs does not make much difference in the utilization rate. The corresponding optimization rate of the total mileage of the routes maintains at a stable state of around 76%, which means the increase in the number of UAVs does not make much difference in the total mileage of the inspection routes. (2) From Figure 10, we can find that the increase in the number of UAVs will lead to the increase in the number of droneports; however, there is no linear positive correlation between them. Moreover, when the number of UAVs is less than 14, α is no less than 3.6, and when the number of UAVs exceeds 18, α is in the range of 2.6 to 3. As the number of UAVs increases, although the multiplier α decreases, the absolute total number of UAVs and distributed droneport increases, which increases the total cost. (3) From Figure 11, the optimization rate of a global inspection cycle increases as the number of UAVs increases. The optimization rate already achieves 75% when the number of the UAVs is only 4, which means the shortest time to complete a global inspection cycle is 5.5 days with 4 UAVs continuously running. (4) We synthesize the multiple indicators of the Panyu district with 888.19 km of inspection roads under different numbers of UAVs in Table 3, where we find that using 4 UAVs has the best cost performance.

Based on above observations, our method can effectively plan the inspection routes, balance the utilization of the UAVs and select the sites for the distributed droneports, which shows great significance for the fully autonomous UAV swarm inspection system for road inspection. Note that the proposed method in this paper remains subject to certain constraints. Specifically, one notable assumption is predicated on the presupposition that each UAV commences its flight with a fully charged battery. As stipulated within the confines of this study, each flight is constrained to a distance ranging from 27 to 30 km. However, it is essential to highlight that if a UAV takes off with a battery charge below full capacity, it may not attain the prescribed flight range of 27 to 30 km. In such instances, the flight route planning must be recalibrated in accordance with the estimated flight range based on the pre-flight battery status. Furthermore, while this paper extensively addresses the intricacies of drone route planning, it regrettably omits the consideration of UAV scheduling. This critical aspect shall be the focal point of our forthcoming research endeavors.

## 7. Conclusions

This paper proposes an inspection route planning method for UAVs that fuses the site selection of the distributed droneports, and evaluates the effectiveness of the proposed method comprehensively. The experimental results show that (1) the proposed method has the capability to plan the inspection routes of the UAVs with a good balance between the total mileage of the routes and the UAV utilization rate; (2) the reuse rate of distributed droneports is significantly improved after optimization; and (3) regardless of the number of drones (within the range of 4–20), the total flight distance of the planned drone routes using the method proposed in this paper changes very little and will not be affected by extremely unreasonable situations. In summary, our method is of great significance for the fully autonomous inspection by the UAV swarm. In the future, we will further consider the retransmission and error detection mechanisms into the optimization model for systematic research.

## Figures and Tables

**Figure 1 sensors-23-08479-f001:**
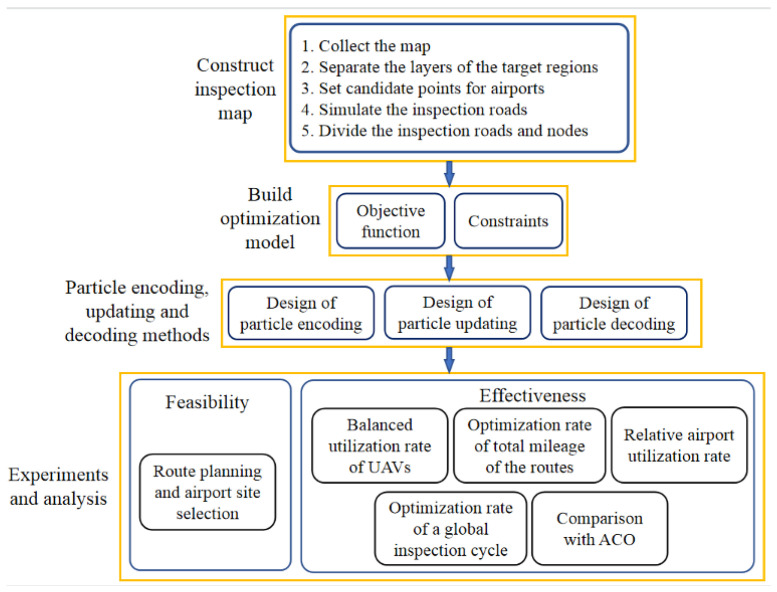
Research Framework.

**Figure 2 sensors-23-08479-f002:**
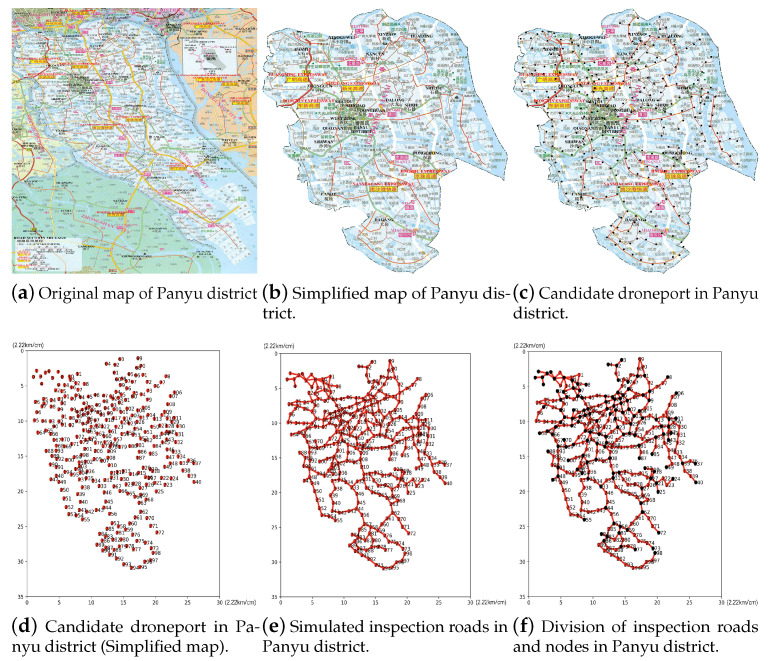
Inspection map construction.

**Figure 3 sensors-23-08479-f003:**
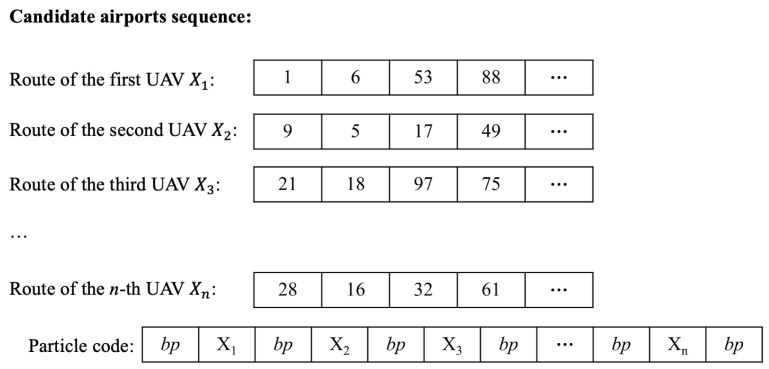
Particle encoding.

**Figure 4 sensors-23-08479-f004:**
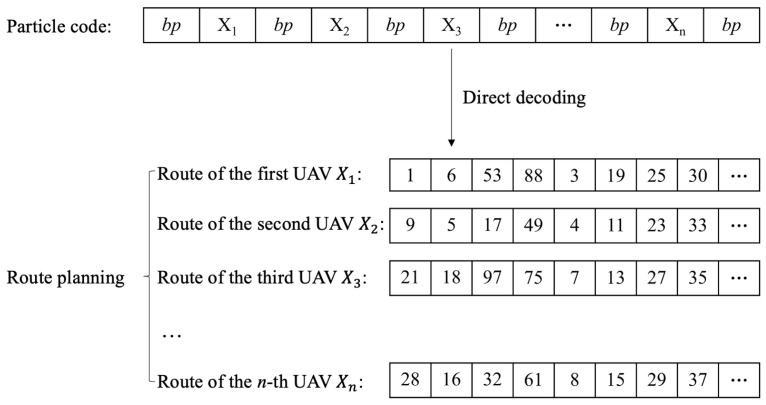
Direct particle decoding.

**Figure 5 sensors-23-08479-f005:**
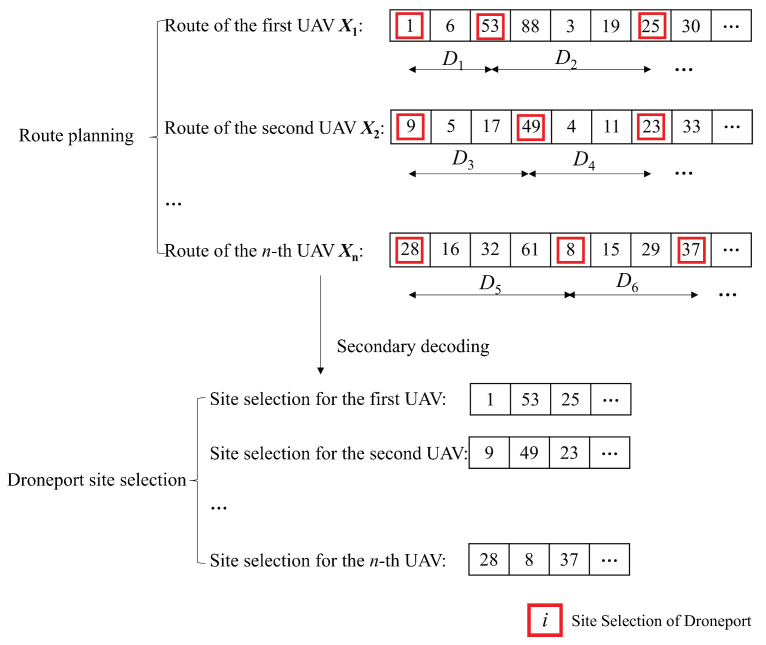
Indirect particle decoding.

**Figure 6 sensors-23-08479-f006:**
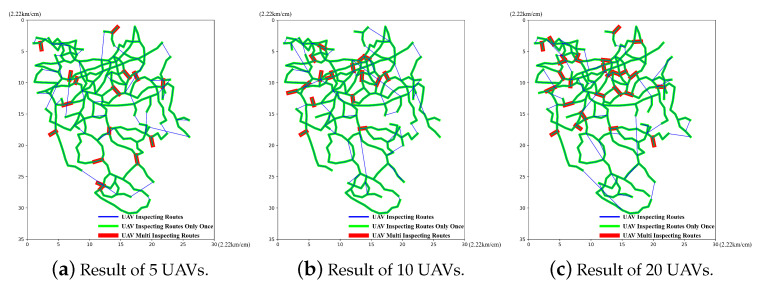
Route planning results.

**Figure 7 sensors-23-08479-f007:**
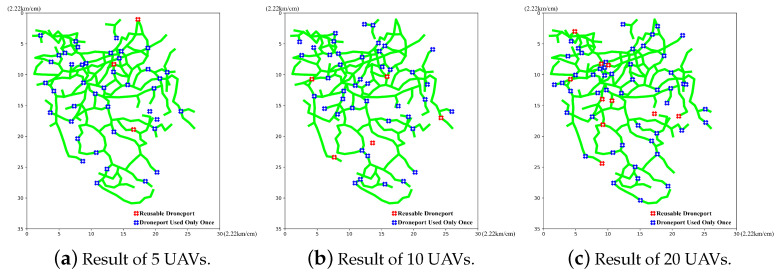
Droneport site selection results.

**Figure 8 sensors-23-08479-f008:**
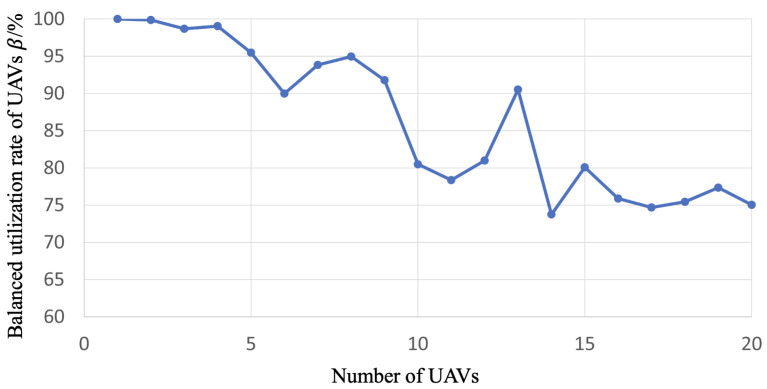
The line chart of the balanced utilization rate of UAVs under different numbers of UAVs.

**Figure 9 sensors-23-08479-f009:**
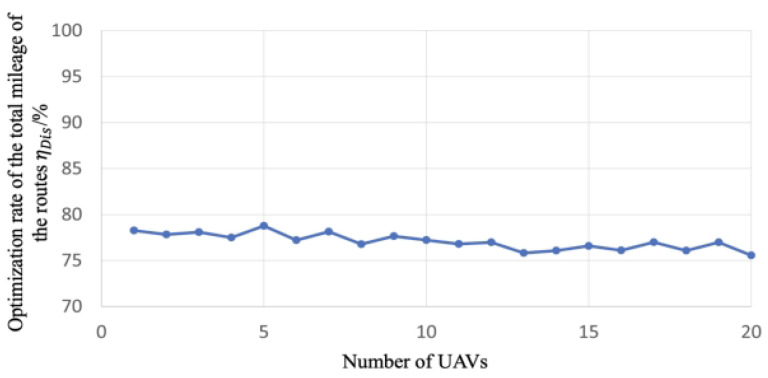
The line chart of the optimization rate of the total mileage of the routes under different number of UAVs.

**Figure 10 sensors-23-08479-f010:**
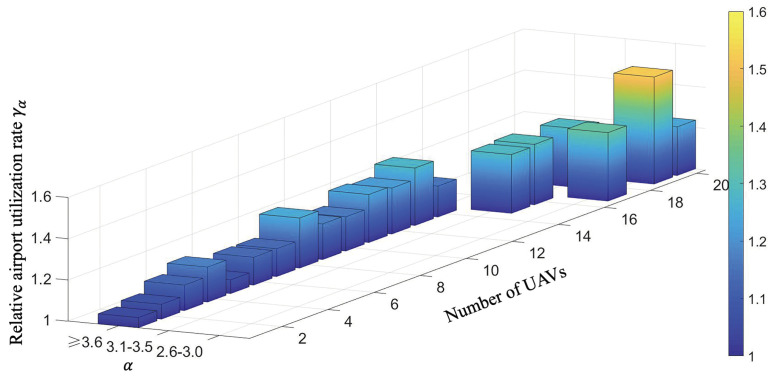
Histogram of relative droneport utilization rate under different numbers of UAVs.

**Figure 11 sensors-23-08479-f011:**
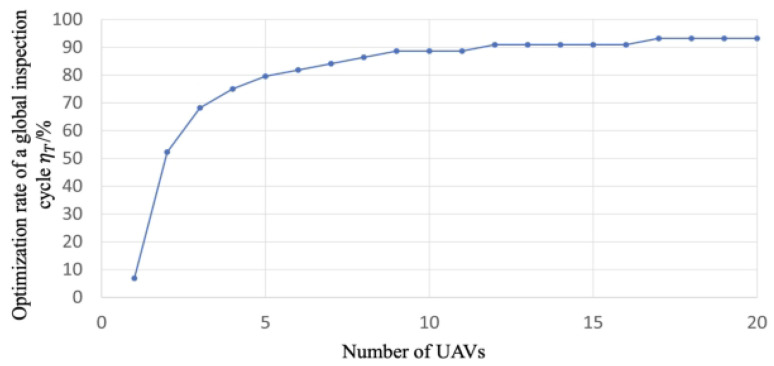
Optimization rate of a global inspection cycle under different numbers of UAVs.

**Table 1 sensors-23-08479-t001:** The tasks and the corresponding reserved time intervals for UAVs during the road inspection process.

Tasks	Reserved Time Intervals
Inspect roads	30 min
Avoid collisions during inspection	10 min
Charge at the droneport	1 h and 15 min
Maintain UAVs (e.g., blow dust, change paddles)	1 h and 15 min

**Table 2 sensors-23-08479-t002:** Comparison with ACO.

Method	Optimization Time/s	Balanced Utilization Rate of UAVs/%	Optimization Rate of the Total Mileage of the Routes/%	Relative Droneport Utilization Rate	Optimization Rate of a Global Inspection Cycle/%
Our method	220.66	90.53	75.82	1.28	90.91
ACO	334.67	69.82	63.43	1.19	88.64

**Table 3 sensors-23-08479-t003:** Comparison of inspection plans with different numbers of UAVs.

Number of UAVs	Balance Utilization Rate (%)	Total Mileage of the Routes (km)	Droneport Reuse Rate (%)	Number of the Droneport	Minimum Time of a Global Inspection (Days)
2	98.69	1141.24	7.3	41	10.5
3	99.03	1137.44	10	40	7
4	99.03	1146.12	17	41	5.5
5	95.48	1127.59	6.8	44	4.5
6	89.98	1150.38	13.6	44	4
7	93.82	1136.68	13.6	44	3.5
8	94.96	1156.79	24.4	45	3

## Data Availability

The original map of Panyu district is available in https://dili.chazidian.com/uploadfile/dili/oldimages/2013/0109/panyu.jpg (accessed on 15 June 2022), and the information of drones used in this manuscript is available in http://www.uflycn.com/html-en/productdetail-53.html (accessed on 28 August 2022).

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
