# Peer review of "A Route Planning Method for UAV Swarm Inspection of Roads Fusing Distributed Droneport Site Selection"

_sensors, 2023, doi:10.3390/s23208479_

Round 1
Reviewer 1 Report
An innovative approach to road inspection that harmonizes UAV utilization and route planning is introduced in this manuscript, and the authors employ roads in the Panyu district of Guangzhou, China, as a case study to validate their optimization strategy. However, several questions should be addressed before it can be accepted.
1. Please clarify the distinction between the 'utilization rate of UAVs' as outlined in the objective function and the 'Balanced Utilization Rate of UAVs' as specified in the evaluation metrics?
2. The 'Overlapped Rate of Inspection Routes' is set as 0.18. What is the rationale behind this particular value?
3. While the authors convincingly demonstrate the feasibility and efficacy of their approach through empirical testing, they do not furnish any quantitative data or performance metrics to substantiate their claims.
Minor editing of English language required.
Reviewer 2 Report
Overall, the study description provided in this thread presents a detailed and comprehensive overview of a UAV-based road inspection system and the associated optimization method. Here are some general and specific comments on the overall study description:
1. The study makes use of mathematical notations and equations. It would be beneficial to provide a glossary or list of symbols for easy reference, especially for readers less familiar with mathematical optimization.
2. While the study mentions the proposed method's strengths, it would be helpful to explicitly address potential weaknesses or limitations of the approach. This can enhance the paper's credibility and guide future research directions.
3. The experimental design covers various aspects of the proposed method's performance. However, a few suggestions for improvement include:
l Elaborate more on the practical implications of the experiments in section 5.2. For instance, why is it important to examine the correlation between the balanced utilization rate of UAVs and the total number of UAVs?
l Consider providing a brief rationale for the choice of the initial population size and the values of learning factors c1 and c2. Explain how these choices might impact the experimental outcomes.
4. The constraint limiting the overlap between inspection routes (0.18 in the presented model) is somewhat arbitrary and may not suit all real-world scenarios. The optimal level of overlap can vary based on specific road network characteristics and inspection goals.
5. Please provide clearer representations of Figure 6 and 7, as the current figures are too small, and the colors used are not easily distinguishable.
6. The method relies on various parameters and constraints, such as the distance between candidate airport points and time intervals for tasks. The performance of the approach may be sensitive to the precise values chosen for these parameters, and finding optimal values can be challenging.
7. The mathematical model presented in the approach is complex and involves multiple objectives and constraints. Solving such a complex optimization problem may require significant computational resources and time, making it less practical for real-time or large-scale applications.
8. Consider changing the term 'airport' to 'droneport' or a similar term to avoid any confusion with conventional airports for legacy aircraft.
Overall, the quality of English in this study description is quite good, but there are areas where improvement is needed. Some sentences are long and complex, which can make the text harder to follow. Additionally, there are some minor grammatical errors that could be refined for better clarity and readability. Overall, the content is informative and understandable, but it could benefit from some editing to enhance its overall quality.
Reviewer 3 Report
A brief summary: In this publication, the authors develop and investigate methods for optimizing the use of UAV swarms for road inspection. Due to the rapid development of UAVs, this area of research is quite relevant and important. The proposed algorithm is quite new and interesting.
General concept comments:
The introduction provides brief descriptions of the results of previous studies, but unfortunately, there is no critical analysis of these results (see Spec. comm.1). The proposed methodology is the strongest point of this paper but requires some minor refinements (see Spec. comm. 2). The results are interesting. The discussion should be supplemented by comparison with other studies and critical analysis (see Spec. comm. 3). The conclusions are logical and clear. The list of references is up-to-date but it could be expanded.
Specific comments
1. I propose to perform a critical analysis of the described studies with an analysis of the main advantages and disadvantages. Such an analysis will allow justifying the purpose and relevance of the study.
2. It is necessary to clarify what technical means (software products) were used to prepare the inspection map. Explain why this map was prepared manually. Because there are a lot of GIS capabilities for preparing such a map.
3. The discussion requires comparison with more methods and approaches. This will help to form the novelty of the study.
